# Sex Difference and Smoking Effect of Lung Cancer Incidence in Asian Population

**DOI:** 10.3390/cancers13010113

**Published:** 2020-12-31

**Authors:** Boyoung Park, Yeol Kim, Jaeho Lee, Nayoung Lee, Seung Hun Jang

**Affiliations:** 1National Cancer Center, Division of Cancer Prevention and Early Detection, National Cancer Control Institute, Goyang 10408, Korea; hayejine@hanyang.ac.kr (B.P.); jhlee214@ncc.re.kr (J.L.); ny2916@ncc.re.kr (N.L.); 2National Cancer Center, Department of Cancer Control and Population Health, Graduate School of Cancer Science and Policy, Goyang 10408, Korea; 3Department of Medicine, Hanyang University College of Medicine, Seoul 04763, Korea; 4Department of Pulmonary, Allergy and Critical Care Medicine, Hallym University Sacred Heart Hospital, Anyang 14068, Korea

**Keywords:** tobacco smoking, lung cancer, sex difference, age-standardized incidence

## Abstract

**Simple Summary:**

This study analyzed the sex difference in the effect of smoking exposure on lung cancer in terms of absolute and relative risks despite the increasing lung cancer incidence in Asian female never smokers. Lung cancer risk is positively associated with the duration of smoking, quantity of smoking, and pack-years of smoking, and negatively associated with the number of years since smoking cessation for both sexes. However, higher lung cancer incidence in men than in women with the same level of smoking exposure was observed, suggesting a higher susceptibility for lung cancer in men. Sex should be considered in combination with smoking history in the selection of a lung cancer screening target population.

**Abstract:**

This study analyzed the sex difference in the effect of smoking exposure on lung cancer in terms of absolute and relative risks despite the increasing lung cancer incidence in Asian female never smokers. A retrospective cohort study was conducted on individuals aged 40–79 years who participated in the national health screening program in 2007 and 2008 with linkage to the Korea Central Cancer Registry records. We evaluated sex differences in the age-standardized incidence rate (ASR) of lung cancer by smoking history and the hazard ratio (HR) after adjusting for potential confounders. ASRs for male and female never smokers were 92.5 and 38.3 per 100,000 person-years, respectively (rate ratio (RR) = 2.4; 95% confidence interval (CI) = 2.3–2.5). ASRs for male and female current smokers with a 30 pack-year smoking history were 305.3 and 188.4 per 100,000 person-years, respectively (RR = 1.6; 95% CI = 1.3–2.0). Smoking was significantly associated with lung cancer risk for both sexes. HRs for former smokers versus never smokers were 1.27 (95% CI = 1.23–1.33) for men and 1.43 (95% CI = 1.16–1.81) for women. HRs for current smokers versus never smokers were 2.71 (95% CI = 2.63–2.79) for men and 2.70 (95% CI = 2.48–2.94) for women. HRs for lung cancer increased similarly in both men and women according to smoking status. However, among Korean individuals with comparable smoking statuses, lung cancer incidence is higher in men than in women. Sex should be considered in combination with smoking history in the selection of a lung cancer screening target population.

## 1. Introduction

Lung cancer is a leading cause of cancer incidence and mortality worldwide [1], and its incidence is higher in men than in women. The age-standardized incidence rate (ASR) for men is approximately 31.5 per 100,000 person-years, which is twice as high as that for women (14.6 per 100,000 person-years) [1]. However, in recent studies, this sex difference in lung cancer incidence has remarkably decreased in developed countries. One suggested reason for this is the changes in smoking habits between sexes, which has been declining in men, but continuously increasing in women [1].

Another trend in lung cancer incidence over the past decades is the increasing number of non-smoking-associated lung cancers, which was more prominent in women previously [2]. One study even proposed that female sex itself is an independent risk factor of lung cancer [3]. Lung cancer in never smokers accounts for 10–25% of all lung cancers, with significant geographical variations [4,5], as high as 30–50% in Southeast and East Asia [6,7]. The continuously increasing incidence of non-smoking-associated lung cancers suggests the importance of risk factors other than smoking in women, especially in these regions. Thus, the question, “Does cigarette smoking influence the development of lung cancer and remain a key risk factor in women as in men in East-Asian population?” has been raised.

To identify whether men and women have different susceptibilities associated with smoking in terms of lung cancer, both relative and absolute risks (incidence rate) need to be considered, but this approach has rarely been adopted. The main objective of this study was to identify sex differences in the effects of smoking on lung cancer. We evaluated sex differences in both the absolute risk in terms of age-standardized incidence rates (ASRs) and relative risk for given exposure levels of smoking. This retrospective cohort study was conducted with the largest Asian population to date.

## 2. Materials and Methods

### 2.1. Study Population and Follow-Up

The Korean National Health Insurance Service (NHIS) conducts biennial health examinations for individuals aged ≥40 years. It also conducts screenings for stomach, liver, colon, breast, and cervix cancers for individuals aged ≥40 years, ≥40 years, ≥50 years, ≥40 years (women only), and ≥20 years (women only), respectively. Participants undergoing health examinations and cancer screening are required to complete a set of questionnaires, in which self-reported information on lifestyle including tobacco smoking, alcohol consumption, leisure time, physical activity, and family history of cancer is collected.

The study population included individuals aged between 40 and 79 years who participated in both health examinations and any of the five cancer screenings in 2007 and 2008. The cancer screening questionnaire included questions on the duration of smoking cessation, which were not excluded in the health examination questionnaires (in the health examination questionnaires, duration of smoking cessation for former smokers was not included).

A total of 6,569,144 participants were included in the study, after excluding 3863 participants who were diagnosed as having lung cancer, 45,952 participants diagnosed as having any other type of cancer before the date of examination, and 30,255 participants diagnosed as having cancer within 6 months after examination. In 6,569,144 participants, person-years were calculated from the date of health examinations until 31 December 2014, date of death, or date of cancer incidence, whichever came first.

We obtained data on cancer incidence by linking the participants’ records to the records in the Korea Central Cancer Registry, which covers nationwide cancer incidence with a completeness of ≥96% [8]. Data on cancer, date of cancer diagnosis, and date of death were obtained by linkage of patient data with Korea Central Cancer Registry records (https://ncc.re.kr/main.ncc?uri=manage02_1) and vital statistics of Statistics Korea (http://kostat.go.kr/portal/eng/pressReleases/8/9/index.board). Cancer sites were classified using the International Classification of Diseases (ICD) 10th version for oncology (https://www.who.int/classifications/icd/), and cancers in the trachea, bronchus, and lung (ICD C33 or C34) were considered as outcomes of lung cancer incidence.

This study was approved by the Institutional Review Board of the National Cancer Center, Korea (Code of approval: NCC20160278, Date: 15 May 2016). De-identified data were obtained from the NHIS, and the requirement for informed consent was waived for this specific study with permission from the Ministry of Health and Welfare and Institutional Review Board of the National Cancer Center. This study was conducted in accordance with the Declaration of Helsinki.

### 2.2. Measurements

Information on smoking habits was available from questionnaires in 2007 and 2008: “Have you smoked more than 100 cigarettes (5 packs) in your lifetime?”; “Do you smoke cigarettes now?”; “How many cigarettes do you smoke per day?” (for current smokers); “When you smoked, how many cigarettes did you smoke?” (for former smokers); “How many years have you smoked in your lifetime?”; and “How many years have passed since you quit smoking?” (for former smokers).

Answers were classified into smoking status (never, former, and current), duration of smoking (1–9 years, 10–19 years, 20–29 years, and ≥30 years), cigarettes smoked per day (1–10 cigarettes, 11–20 cigarettes, and ≥20 cigarettes), and pack-years (years of smoking × cigarettes smoked per day/20, classified as 0–9 pack-years, 10–19 pack-years, 20–29 pack-years, and ≥30 pack-years or 0–19 pack-years and ≥20 pack-years). For former smokers, data on years since smoking cessation were classified as 0–4 years and ≥5 pack-years since cessation.

Number of days of alcohol consumption per week during the last 1 year (none, 1–2, ≥3), number of days of sweating exercise per week (none, 1–2, ≥3), and family history of cancer (no, yes) were considered as covariates to be adjusted for; these data were obtained through questionnaires. Moreover, body mass index based on height and weight during health examinations (<18.5 kg/m^2^, 18.5–24.9 kg/m^2^, ≥25 kg/m^2^) and data on the history of chronic obstructive pulmonary disease, pneumoconiosis, and interstitial pulmonary disease (yes and no) from health insurance claims data in NHIS were included as adjusted covariates.

### 2.3. Statistical Analysis

The ASR of lung cancer per 100,000 person-years and the 95% confidence interval (CI) were calculated for each subgroup of age (5-year bands), sex, and smoking status, projected in the mid-year population of the year 2000. We also calculated the rate ratio (RR) of ASR of lung cancer between men and women.

The hazard ratio (HR) and 95% CI of smoking status, smoking duration, quantity of smoking, and pack-years were calculated using Cox proportional hazards regression. The effects were also stratified by smoking status (former and current smokers). For former smokers, the HRs of years since cessation, combination of pack-years, and years since cessation were also estimated. All analyses were carried out by stratifying sex. The interaction effect between sex and each smoking variable on lung cancer risk was analyzed using Cox regression models with the interaction term. We conducted the same analysis for all populations (men and women), with female never smokers as the reference. All analyses were adjusted for age as the continuous variable and alcohol consumption; physical activity; body mass index; family history of cancer; and medical history of lung diseases, including emphysema, chronic pulmonary obstructive disease, pneumoconiosis, and interstitial pulmonary disease, as categorical variables. If there was missing information on adjusted variables, the variables were treated as dummy variables. As a sensitivity analysis, we did same analysis including people having any other types of cancer except lung cancer before the date of examination. All analyses were carried out using SAS version 9.4 (SAS Institute Inc., Cary, NC, USA).

## 3. Results

### 3.1. Characteristics

The baseline characteristics of the male and female individuals of the study population are presented in Table 1. The mean age of the men and women was 55.4 and 54.8 years, respectively, at enrollment. The proportion of individuals who did not drink alcohol and did not perform sweating exercise was higher among women (42.5% and 33.1% in men and 79.9% and 42.3% in women, respectively). The proportions of never, former, and current smokers were 47.3%, 21.0%, and 31.7% in men and 97.3%, 0.6%, and 2.1% in women, respectively. In 43,525,986 person-years of follow-up (mean 6.6 years), 33,570 individuals (22,423 men and 11,147 women) were diagnosed as having trachea, bronchus, or lung cancer.

### 3.2. Sex Difference in the Incidence of Lung Cancer

Table 2 shows the crude incidence rate and age-standardized incidence rate (ASR) of lung cancer in men and women as well as the RR between sexes. Among never smokers, the ASR was 92.5 (95% CI = 0.5–94.6) in men and 38.3 (95% CI = 37.5–39.1) in women per 100,000 person-years, with an RR of 2.4 (95% CI = 2.3–2.5). Among former smokers with a smoking history of 0–19 pack-years, the RR between men and women did not show statistically significant differences, but former smokers with a ≥20 pack-year smoking history showed an RR of 1.7 (95% CI = 1.1–2.5), suggesting a significantly higher incidence in male former smokers than in female former smokers given the same smoking quantity. The RR between male and female current smokers ranged between 1.6 and 1.8 in all smoking quantity strata, and all were statistically significant. In current smokers with a ≥30 pack-year smoking history, the crude and standardized incidence rates were 350.9 and 305.3, respectively, in men and 320.5 and 188.4, respectively, in women per 100,000 person-years. Figure 1 and Figure 2 present cumulative lung cancer incidence in men and women according to smoking status and pack-years.

### 3.3. Sex Difference in the Effect of Smoking Status on Lung Cancer

Table 3 shows the sex differences in the effects of smoking on the risk of lung cancer as HRs. The HRs of lung cancer incidence for female former and current smokers compared with female never smokers were 1.43 (95% CI = 1.16–1.81) and 2.70 (95% CI = 2.48–2.94), respectively. The HRs were 1.27 (95% CI = 1.23–1.33) and 2.71 (95% CI = 2.63–2.79) for male former and current smokers, respectively. There was no statistically significant interaction between sex and smoking status on lung cancer risk (*P*-interaction: 0.261).

The HR of lung cancer incidence for male never smokers compared with female never smokers was 2.41 (95% CI = 2.34–2.48). The risk of lung cancer in current male smokers was 6.27 times higher (95% CI = 6.08–6.47) than that in female never smokers (Appendix A).

### 3.4. Sex Difference in the Effect of Detailed Smoking History on Lung Cancer

In terms of smoking duration, a significant interaction was observed (*P*-interaction < 0.05). The dose–response relationship was prominent in women. However, in men, only those who had smoked for ≥20 years or ≥30 years showed a statistically significant increase in lung cancer incidence. Daily smoking quantity was positively associated with the risk of lung cancer in both sexes, but we found no interaction between sex and daily smoking quantity. The HR of male current smokers smoking >20 cigarettes/day was 2.18 (95% CI = 2.03–2.36) and that of female current smokers smoking >20 cigarettes/day was 1.74 (95% CI = 1.39–2.19), compared with those who smoked 1–10 cigarettes/day. Despite the increasing trend, we did not observe a statistically significant association between quantity of smoking and lung cancer risk in female former smokers because of the small number of cases.

We also observed a positive association between pack-year smoking history and lung cancer incidence. The HRs for lung cancer incidence among all male and female smokers with a ≥30 pack-year smoking history, compared with smokers with a 0–9 pack-year smoking history, were 2.91 (95% CI = 2.69–3.15) and 2.59 (95% CI = 2.05–3.28), respectively.

### 3.5. Sex Difference in the Effect of Smoking Cessation on Lung Cancer

We observed a negative relationship between the duration of smoking abstinence and lung cancer incidence. The lung cancer risk significantly decreased in both sexes if smoking cessation was ≥5 years. The HRs of lung cancer incidence for ≥5 years of smoking cessation were 0.46 (95% CI = 0.33–0.64) in women and 0.38 (95% CI = 0.36–0.40) in men. However, the HR in male former smokers with ≥5 years of smoking cessation compared with female current smokers was 0.92 (95% CI = 0.84–1.01). This shows that, although smoking cessation rapidly decreased the risk of lung cancer, male former smokers still had a similar risk to that of female current smokers (Appendix A). Figure 3 presents cumulative lung cancer incidence according to years since cessation in former smokers.

The HRs of lung cancer in combined strata of smoking quantity and duration of smoking abstinence are summarized in Table 4. Each HR was significantly <1.0 in all strata except for former female smokers with a >10 pack-year smoking history. The HRs of former smokers with a <10 pack-year smoking history and a duration of smoking abstinence ≥5 years were 0.24 (95% CI = 0.21–0.27) in men and 0.31 (95% CI = 0.18–0.51) in women, and the HRs of former smokers with a ≥30 pack-year smoking history and a duration of smoking abstinence <5 years were 0.87 (95% CI = 0.81–0.93) in men and 0.79 (95% CI = 0.37–1.67) in women. The HRs of male former smokers with a ≥30 pack-year smoking history (regardless of the duration of smoking abstinence) or a ≥20 and <30 pack-year smoking history and a duration of smoking abstinence <5 years were still 1.34–2.09 times higher than that for female current smokers, which was a significant difference (Appendix A).

The sensitivity analysis including people with other than lung cancer before the date of examination showed comparable results.

## 4. Discussion

In this study, we found higher ASRs of lung cancer in men with equal smoking exposures, especially in never smokers than women. The RRs of ASR between men and women were 2.4 and 1.6 for never smokers and current smokers with a ≥30 pack-year smoking history, respectively. Our results suggest a higher absolute risk of lung cancer in men than in women, irrespective of smoking exposures.

Our results contradict previously reported results on sex differences in lung cancer incidence in the Western population, but are consistent with the results in the East Asian population. Recent prospective studies on the Western population consistently reveal a higher incidence of lung cancer in female never smokers than in male never smokers [2,9,10,11,12]. Higher exposures to cooking oil fume, secondhand smoking, and human papilloma virus were the suggested causative factors [4,7,13]. However, the results in the East Asian population were opposite. The lung cancer incidence in female never smokers was lower than that in male never smokers in Japan and Korea [14,15], although the proportion of female never smokers who were diagnosed with lung cancers in these regions was higher than that in Western countries [13]. A lower proportion of smoking among women with lung cancer [6] and a higher proportion of non-smokers among Asian women [16] possibly led to the lower incidence rate of lung cancer in female never smokers, especially in East Asia.

The sex difference in lung cancer incidence was greater in non-smokers than in smokers. The magnitude of the RR of ASR between men and women was higher in non-smokers than in smokers. Higher occupational exposure in men, such as exposure to asbestos because of asbestos mining until the mid-1980s or wide asbestos use in manufacturing (completely banned in 2009) [17], is one of the possible reasons for the higher lung cancer incidence in male non-smokers, and hence the higher RR of ASR between men and women. Furthermore, exposure to secondhand smoking was much higher in men than in women because of the extremely high smoking rate (approximately 70%) in the Korean male population in 1970–1990. We also observed a similar trend in the sex difference among former and current smokers, which is consistent with the results from previous studies [9,11,12,18], but inconsistent with those of studies in early 2000s, which showed a similar risk of lung cancer incidence between sexes [10].

Although lung cancer incidence has significantly increased with an increase in smoking consumption history, we observed no interactive effect of sex and smoking exposure on HR in general, except for duration of smoking and pack-years for former smokers. The lack of an interaction between sex and smoking exposure is consistent with the results from recent cohort studies, whose results are still debatable [9,19]. For example, it is suggested that one possible reason for the lack of an interaction effect between sex and smoking exposure is the categorical classification of variables. A significant interaction effect between sex and pack-years, smoking quantity, and duration of smoking was observed for continuous variables [11].

However, our results show that the sex difference was more prominent for the effect of smoking duration on lung cancer risk. Women might be more susceptible to duration of exposure than men. This is supported by the results that direct exposure effects of a longer duration of exposure was observed at lower smoking intensities (excess odds ratio per pack-year increased with intensity), but an inverse exposure effect of a longer duration of exposure was observed at higher smoking intensities (excess odds ratio per pack-year decreased with intensity) [20]. Considering that the quantity of smoking, represented as the number of cigarettes per day, was lower in women than in men, with a particularly higher difference between sexes in Asian individuals [21], a finding of a more direct effect of smoking duration in women in this study would be reasonable.

The number of years since cessation was negatively associated with lung cancer risk, as found previously [22]. We also assessed the combined effect of pack-years and number of years since smoking cessation on lung cancer risk. Lung cancer risk was generally reduced when both pack-years and cessation years were lower, as expected. We still observed a lower lung cancer risk in men who had a >30 pack-year smoking history, but had quit smoking <5 years ago than in male current smokers. However, in female former smokers, we did not observe a statistically significant combined effect of pack-years and cessation years among those with a ≥10 pack-year smoking history. We believe that the sample size in these categories was too small to verify the results. We observed a significant effect of the number of years since cessation on lung cancer risk in female former smokers when considered alone (Table 3).

The difference in HR between former or current smokers and never smokers was smaller than those in previous studies in both Western [9,11,12,18,23,24,25] and Asian countries [14,26]. However, both the lower effect of smoking on lung cancer and the higher mortality rate from lung cancer observed in Asian countries suggest that the results of our study are still meaningful. A recent meta-analysis of studies in the Korean population also showed similar pooled RRs (2.58 and 2.37 in male and female current smokers, respectively) [27]. The shorter follow-up period in this study (maximum of 8 years) and the higher baseline incidence rates in never smokers could also have led to the smaller difference in HR between former or current smokers and never smokers. It is also possible that the current lung cancer incidence rate does not fully reflect the risk of smoking, as the tobacco epidemic in Asian countries has reached a peak [27].

Another interesting observation from this study was that both male never smokers and male former smokers who had quit smoking >5 years prior still showed a comparable lung cancer risk to that of female current smokers. This result reflects a higher baseline risk in male never smokers. The risk of lung cancer was six times higher in male current smokers than in female never smokers. Studies have suggested that a lower smoking prevalence in women or a difference in the epidemiologic smoking trend between sexes may be attributed to the lower lung cancer incidence in women. For example, it is suggested that the lung cancer incidence has reached its peak in men and started to decrease recently, but it is yet to reach its peak in women [1]. In Korea, the lung cancer incidence has decreased in both men and women since 2005 and 2011, respectively [28], despite the fact that the prevalence of tobacco smoking has decreased in men since 2012 and has been stable in women [29]. Such a trend could not be explained by the “maturity of tobacco epidemic”—a decrease in lung cancer incidence followed by a highest/maximum smoking rate—which has been applied to explain regional and sex differences in lung cancer incidence and trend [1]. Future studies shall further evaluate the potential cause of the low baseline lung cancer risk in women.

A few limitations of this study should be mentioned. First, the study population may not represent the general Korean population because the study population consists of screening examinees. Several studies have shown that participants of cancer screening have a better socioeconomic status and better health behaviors than do non-participants [30,31]. The current smoking rate in men and women in this study population was lower than that in the general Korean adult population [29,32]. However, considering the participation rate of the national health screening program (≥70%), the generalizability of their results could be applied to most of the Korean population. Second, it is possible that some current smokers in 2007 and 2008 may have quit smoking during the follow-up period, but we did not take this possibility into consideration. Third, we could not consider the histological types of lung cancer because of the lack of information, but previous studies have shown that smoking itself consistently increased the risk of lung cancer, irrespective of histological type [11,33]. Fourth, the information on smoking was obtained from a self-administered questionnaire that was pre-submitted for health examination, which could have caused information bias. Women who smoke are still “frowned upon” in some Asian countries; therefore, we cannot completely exclude the possibility of overreported female never smokers or underreported female former or current smokers. In this study, the lung cancer incidence in non-smokers was extremely low, and the RR of ASR between men and women was higher in non-smokers than in smokers, suggesting a much lower lung cancer incidence in female non-smokers than in male non-smokers. Thus, underestimation of the association of smoking and lung cancer in women because of underreporting is unlikely. Finally, passive smoking and occupational and environmental exposures such as asbestos or radon exposures were not considered in the data set. Products containing asbestos have been prohibited partly since 1997 and completely since 2009 [17], and the proportion of workers exposed to asbestos was small (<1%) [34]. The radon concentration in residential area was half the standard of Korea and the United States [35]. Thus, we consider that occupational and environmental exposures would be very low in the population. However, to adjust for the effect of occupational and environmental exposures, we adjusted for the lung diseases as surrogate confounders that could be caused by occupational and environmental exposures. In addition, other confounding variables such as cooking habits or diet could not be adjusted for because of unavailable information.

The strengths of this study are as follows. First, this study was conducted in a large Korean population with the retrospective cohort approach using a nationwide cancer registry. This allowed almost complete follow-up. Second, to the best of the authors’ knowledge, this is the first study that considered the combined effect of pack-years and number of years since smoking cessation on lung cancer risk for former smokers.

## 5. Conclusions

Therefore, we found a higher lung cancer incidence in men than in women with the same level of smoking exposure, suggesting a higher susceptibility for lung cancer in men. Lung cancer risk is positively associated with the duration of smoking and negatively associated with the number of years since smoking cessation according to the HRs for both sexes. Our results highlight the importance of sex difference in the smoking effects on lung cancer, at least in East Asia. Our results imply that sex should be considered in combination with smoking history in the selection of a lung cancer screening target population.

## Figures and Tables

**Figure 1 cancers-13-00113-f001:**
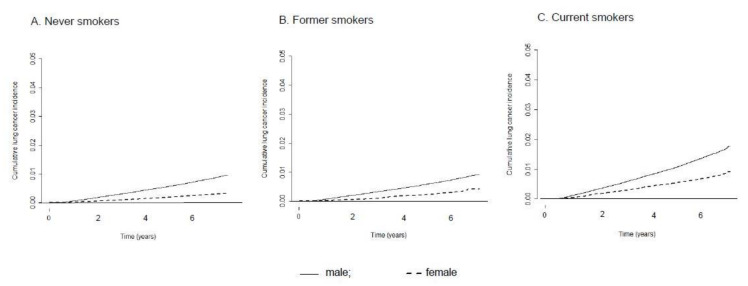
Cumulative lung cancer incidence in men and women according to smoking status. (**A**) Never smokers, (**B**) Former smokers and (**C**) Current smokers.

**Figure 2 cancers-13-00113-f002:**
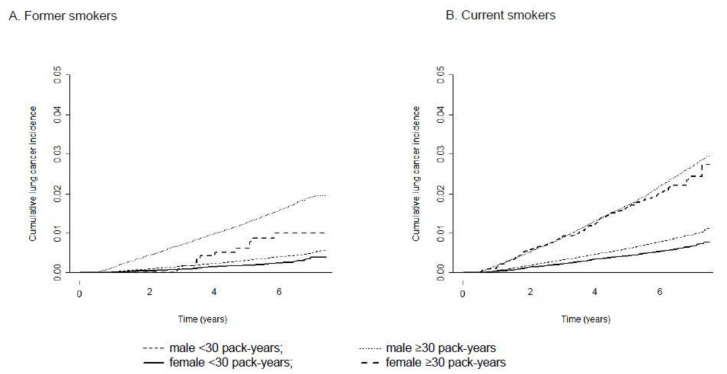
Cumulative lung cancer incidence in men and women according to smoking status and pack-years. (**A**) Former smokers and (**B**) Current smokers.

**Figure 3 cancers-13-00113-f003:**
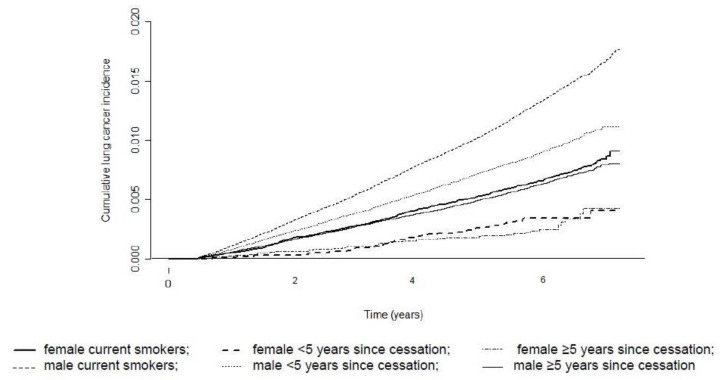
Cumulative lung cancer incidence according to years since cessation in former smokers compared with current smokers.

**Table 1 cancers-13-00113-t001:** Baseline characteristics of the study population.

Characteristics	Male	Female
Age, mean (SD)	55.4 (10.3)	54.8 (10.2)
Number of days of alcohol consumption		
None	1,039,557 (42.5)	3,294,571 (79.9)
1–2/week	708,169 (28.9)	396,165 (9.6)
≥3/week	622,680 (25.4)	113,522 (2.8)
Missing	77,530 (3.2)	316,950 (7.7)
Number of days of sweating exercise		
None	809,073 (33.1)	1,742,508 (42.3)
1–2/week	812,531 (33.2)	1,166,595 (28.3)
≥3/week	815,661 (33.3)	1,198,527 (29.1)
Missing	10,671 (0.4)	13,578 (0.3)
Body mass index		
<18.5 kg/m^2^	36,558 (1.5)	61,006 (1.5)
18.5–24.9 kg/m^2^	1,055,025 (43.1)	1,945,352 (47.2)
≥25 kg/m^2^	657,009 (26.8)	1,039,464 (25.2)
Missing	699,344 (28.6)	1,075,386 (26.1)
Family history of cancer		
No	1,991,495 (81.4)	3,281,754 (79.6)
Yes	456,441 (18.7)	839,454 (20.4)
Missing	-	-
History of chronic pulmonary obstructive disease		
No	2,379,348 (97.2)	4,052,245 (98.3)
Yes	68,588 (2.8)	68,963 (1.7)
History of pneumoconiosis		
No	2,446,322 (99.9)	4,120,887 (100)
Yes	1614 (0.1)	321 (<0.1)
History of interstitial pulmonary disease		
No	2,445,434 (99.9)	4,118,519 (99.9)
Yes	2502 (0.1)	2689 (0.1)

**Table 2 cancers-13-00113-t002:** Incidence of lung cancer by sex and smoking status at baseline.

	Male						Female					
	PkY	N	*N* of PY	*N* of Events	CR ^1^	ASR ^2^	N	N of PY	N of Events	CR ^1^	ASR ^2^	RR ^3^ (95% CI) ^3^
Total		244,7936	15,939,310	22,423	140.7	134.0 (132.2–135.8)	4,121,208	27,586,676	11,147	40.4	39.7 (39.0–40.5)	3.4 (3.3–3.4)
Never		115,7106	7,552,011	8561	113.4	92.5 (90.5–94.6)	4,010,591	26,855,434	10,500	39.1	38.3 (37.5–39.1)	2.4 (2.3–2.5)
Past	0–19	277,398	1,819,539	904	49.7	64.7 (59.9–69.6)	23,250	153,742	52	33.8	50.3 (36.0–64.6)	1.3 (0.9–1.7)
	≥20	236,362	1,528,950	2727	178.4	144.7 (138.7–150.7)	2861	18,723	26	138.9	87.6 (51.5–123.6)	1.7 (1.1–2.5)
Current	0–9	68,039	444,815	377	84.8	145.1 (128.9–161.2)	45,751	303,737	156	51.4	81.4 (67.9–94.8)	1.8 (1.5–2.2)
	10–19	203,575	1,328,750	1434	107.9	185.1 (174.7–195.5)	24,063	158,746	200	126.0	118.0 (101.2–134.9)	1.6 (1.3–1.8)
	20–29	235,963	1,534,630	2347	152.9	219.7 (210.1–229.3)	9188	60,411	98	162.2	126.2 (99.8–152.6)	1.7 (1.4–2.1)
	≥30	269,493	1,730,615	6073	350.9	305.3 (296.3–314.3)	5504	35,883	115	320.5	188.4 (151.5–225.4)	1.6 (1.3–2.0)

PkY, pack-years; N, number; PY, person-years; RR, rate ratio; CI, confidence interval; ^1^ CR, crude incidence rate per 100,000 person-years; ^2^ ASR, age-standardized incidence rate per 100,000 person-years. The standard population was 2000 mid-year population aged 40–79 years of Korea ^3^ RR, rate ratio between men and women.

**Table 3 cancers-13-00113-t003:** Adjusted risk of lung cancer according to the measures of various smoking statuses.

Smoking Measures		Male			Female		Sex–Smoking Interaction
	*N*	Adjusted HR (95% CI) ^1^	*p*-Value	*N*	Adjusted HR (95% CI) ^1^	*p*-Value	
Smoking status							
Never	1,157,106	1.00 (reference)	-	4,010,591	1.00 (reference)	-	-
Former	513,760	1.27 (1.23–1.33)	<0.001	26,111	1.43 (1.16–1.81)	<0.001	0.261
Current	777,070	2.71 (2.63–2.79)	<0.001	84,506	2.70 (2.48–2.94)	<0.001	-
Duration of smoking							
1–9 years	53,677	1.00 (reference)	-	25,164	1.00 (reference)	-	-
10–19 years	186,724	0.95 (0.79–1.14)	0.558	34,884	1.40 (0.95–2.06)	0.093	-
20–29 years	475,970	1.48 (1.26–1.75)	<0.001	29,250	2.09 (1.45–3.01)	<0.001	<0.001
≥30 years	574,023	3.07 (2.62–3.61)	<0.001	21,269	3.16 (2.20–4.54)	<0.001	-
Duration of former smoking							
1–9 years	41,386	1.00 (reference)	-	9643	1.00 (reference)	-	-
10–19 years	129,515	1.09 (0.85–1.38)	0.496	8331	2.39 (0.85–6.73)	0.099	-
20–29 years	169,258	1.52 (1.22–1.90)	<0.001	4849	5.49 (2.07–14.56)	<0.001	0.017
≥30 years	173471	2.81 (2.27–3.49)	<0.001	3265	5.92 (2.17–16.15)	<0.001	-
Duration of current smoking							
1–9 years	12,291	1.00 (reference)	-	15,521	1.00 (reference)	-	-
10–19 years	57,209	0.81 (0.60–1.08)	0.151	26,553	1.14 (0.75–1.73)	0.538	-
20–29 years	306,712	1.08 (0.85–1.39)	0.534	24,401	1.52 (1.02–2.25)	0.038	0.019
≥30 years	400,552	1.81 (1.42–2.31)	<0.001	18,004	2.36 (1.60–3.48)	<0.001	-
Quantity of smoking							
1–10 cigarettes/day	108,145	1.00 (reference)	-	39,841	1.00 (reference)	-	-
11–20 cigarettes/day	491,187	1.39 (1.30–1.50)	<0.001	49,727	1.37 (1.14–1.65)	<0.001	0.143
>20 cigarettes/day	691,498	2.03 (1.89–2.17)	<0.001	21,049	1.69 (1.36–2.09)	<0.001	-
Quantity of smoking in former smokers							
1–10 cigarettes/day	48,946	1.00 (reference)	-	11,361	1.00 (reference)	-	-
11–20 cigarettes/day	200,661	1.43 (1.23–1.66)	<0.001	10,334	1.56 (0.92–2.64)	0.102	0.180
>20 cigarettes/day	264,153	2.17 (1.88–2.51)	<0.001	4416	1.49 (0.80–2.77)	0.204	
Quantity of smoking in current smokers							
1–10 cigarettes/day	59,199	1.00 (reference)	-	28,480	1.00 (reference)	-	-
11–20 cigarettes/day	290,526	1.43 (1.32–1.55)	<0.001	39,393	1.33 (1.09–1.62)	0.005	0.208
>20 cigarettes/day	427,345	2.18 (2.03–2.36)	<0.001	16,633	1.74 (1.39–2.19)	<0.001	-
Pack-year							
0–9	186,132	1.00 (reference)	-	64,184	1.00 (reference)	-	-
10–19	362,880	1.40 (1.28–1.53)	<0.001	28,880	1.70 (1.40–2.08)	<0.001	-
20–29	345,016	1.91 (1.76–2.07)	<0.001	10,844	1.89 (1.45–2.41)	<0.001	0.470
≥30	396,802	2.91 (2.69–3.15)	<0.001	6709	2.59 (2.05–3.28)	<0.001	-
Pack-year in former smokers							
0–9	118,093	1.00 (reference)	-	18,433	1.00 (reference)	-	-
10–19	159,305	1.24 (1.08–1.42)	<0.001	4817	2.37 (1.35–4.14)	<0.001	-
20–29	109,053	1.75 (1.53–2.00)	<0.001	1656	3.15 (1.62–6.10)	<0.001	0.026
≥30	127,309	2.80 (2.48–3.16)	<0.001	1205	2.26 (1.08–4.76)	0.042	-
Pack-year in current smokers							
0–9	68,039	1.00 (reference)	-	45,751	1.00 (reference)	-	-
10–19	203,575	1.24 (1.10–1.39)	<0.001	24,063	1.54 (1.24–1.90)	<0.001	
20–29	235,963	1.50 (1.34–1.67)	<0.001	9188	1.66 (1.29–2.15)	<0.001	0.379
≥30	269,493	2.29 (2.06–2.54)	<0.001	5504	2.52 (1.96–3.23)	<0.001	
Years since cessation in former smokers							
Current smoker	777,070	1.00 (reference)	-	84,506	1.00 (reference)	-	-
0–4	156,560	0.69 (0.65–0.73)	<0.001	11,981	0.65 (0.47–0.90)	0.009	0.469
≥5	357,200	0.38 (0.36–0.40)	<0.001	14,130	0.46 (0.33–0.64)	<0.001	-

*N*, number; HR, hazard ratio; CI, confidence interval. ^1^ Adjusted for age; alcohol consumption; physical activity; body mass index; family history of cancer; and medical history of lung diseases including emphysema, chronic pulmonary obstructive disease, pneumoconiosis, and interstitial pulmonary disease.

**Table 4 cancers-13-00113-t004:** Adjusted risk of lung cancer according to combination of smoking quantity and duration of smoking abstinence.

	PkY	Years Since Cessation	Male			Female		
			*N*	Adjusted HR (95% CI) ^1^	*p*-Value	*N*	Adjusted HR (95% CI) ^1^	*p*-Value
Current	-	-	1,157,106	1.00 (reference)	-	4,010,591	1.00 (reference)	-
Past	<10	<5	20,997	0.34 (0.26–0.45)	<0.001	7845	0.34 (0.18–0.63)	0.001
-	-	≥5	97,096	0.24 (0.21–0.27)	<0.001	10,588	0.31 (0.18–0.51)	<0.001
-	<20	<5	44,599	0.47 (0.41–0.55)	<0.001	2530	0.83 (0.50–1.38)	0.469
-	-	≥5	114,706	0.26 (0.24–0.29)	<0.001	2287	0.60 (0.34–1.06)	0.080
-	<30	<5	39,048	0.57 (0.51–0.65)	<0.001	920	1.00 (0.50–2.02)	0.994
-	-	≥5	70,005	0.37 (0.34–0.41)	<0.001	736	0.84 (0.40–1.78)	0.649
-	≥30	<5	51,916	0.87 (0.81–0.93)	<0.001	686	0.79 (0.37–1.67)	0.535
-	-	≥5	75,393	0.55 (0.52–0.59)	<0.001	519	0.47 (0.18–1.26)	0.135

PkY, pack-years; *N*, number; HR, hazard ratio; CI, confidence interval. ^1^ Adjusted for age; alcohol consumption; physical activity; body mass index; family history of cancer; and medical history of lung diseases including emphysema, chronic pulmonary obstructive disease, pneumoconiosis, and interstitial pulmonary disease.

## Data Availability

The data presented in this study are available on request from the NHIS. The data are not publicly available due to Personal Information Protection Act.

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
