# Peer review of "Sex Difference and Smoking Effect of Lung Cancer Incidence in Asian Population"

_cancers, 2020, doi:10.3390/cancers13010113_

Round 1
Reviewer 1 Report
This is a retrospective cohort study of cancer incidence in an Asian population. The authors are focused on differences of the effects of smoking, including quitting smoking, by sex. The researchers studied a large population of men and women, ages 40-79, and consisted of those who participated in national health screening program. They followed the cohort for eight years and used national registry data to gather incident cases of lung, bronchus and trachea cancers (examined all together). Exposure data were gathered via surveys.
They performed summary statistics and Cox regression modeling. Cox modeling allowed for adjustment by potential confounders (though important confounders could not be measured and included – see below). The main message seemed to appear in both the age-standardized rates as well as the Cox modeling.
The authors conclude that cancer rates were higher in men than in women. This was shown in age-adjusted rate comparisons and in hazard ratios.
This population was likely wealthier than the general population; the authors note this. However, I have some concerns the authors might address in greater detail.
It is possible that this population of women smokes less than those from lower SES. I assume this population is also less likely to face additional environmental and occupational exposures that might produce confounding. How does this affect generalizability of their results?
The authors note that they could not control for occupational and environmental exposures (particularly asbestos and radon) and that, instead, they adjusted for presence of lung diseases that could be caused by occupational and environmental exposures. I am not familiar with circumstances in Korea, but occupational and environmental exposures commonly vary by gender (the main focus of this study), SES (this sample is of higher SES than the general Korean population), and even smoking status. The authors do point out some potential for confounding but their concluding statement about this (lines 305-06) – that they adjusted for lung diseases associated with occupational and environmental exposures – does not feel to me a sufficient acknowledgement of the potential for confounding. Lung disease was reported in about 3% of men and 2% of women. This does not likely reflect adequate measurement of these other exposures of concern. How much of their findings do the authors think might be attributable to confounding?
Given they mention geographic variation in cancer rates by gender, their question ““Does cigarette smoking influence the development 53 of lung cancer and remain a key risk factor in women as in men?” has been raised on lines 53-54 seems like it should have a geographical component.
One statement which appears twice (in the Simple Summary, lines 17-19, and Conclusions, lines 312-13) is confusing to this Reviewer: “However, higher lung cancer incidence in men than in women with the same level of smoking exposures was observed, suggesting a higher susceptibility for lung cancer in women.” Doesn’t this higher rate among men at the same level of exposure reflect greater susceptibility in men?
There appears to be no mention of the figures until the figures actually appear at the end of the manuscript. The figures should either be presented explicitly in the body of the paper or be removed.
The authors provide p-values for interaction terms between sex and smoking variables (for example on lines 160-61) but seem to do sex-stratified analyses throughout. How did the authors produce and evaluate interaction effects?
Another confusing statement is found on line 221-23: “A lower proportion of smoking among women with lung cancer (lower numerator)6 and a higher proportion of non-smokers among Asian women (higher denominator)16 possibly led to the lower incidence rate of lung cancer in female never smokers, especially in East Asia.” I think I understood their point given the sentence preceding this one, but it is difficult to follow. The authors talk about proportions in numerators and denominators instead of numbers which is confusing to people who expect numerators to be numbers and denominators to be person-time. I suggest rewording this sentence for clarity.
Line 295 “showen” should be “shown.”
Author Response
Responses to the Reviewer 1’s Comments
Thank you for the valuable comments.
We have revised the paper in accordance with the comments, and the revisions are summarized below and highlighted in the manuscript.
This is a retrospective cohort study of cancer incidence in an Asian population. The authors are focused on differences of the effects of smoking, including quitting smoking, by sex. The researchers studied a large population of men and women, ages 40-79, and consisted of those who participated in national health screening program. They followed the cohort for eight years and used national registry data to gather incident cases of lung, bronchus and trachea cancers (examined all together). Exposure data were gathered via surveys.
They performed summary statistics and Cox regression modeling. Cox modeling allowed for adjustment by potential confounders (though important confounders could not be measured and included – see below). The main message seemed to appear in both the age-standardized rates as well as the Cox modeling.
The authors conclude that cancer rates were higher in men than in women. This was shown in age-adjusted rate comparisons and in hazard ratios.
This population was likely wealthier than the general population; the authors note this. However, I have some concerns the authors might address in greater detail.
It is possible that this population of women smokes less than those from lower SES. I assume this population is also less likely to face additional environmental and occupational exposures that might produce confounding. How does this affect generalizability of their results?
Response: Thank you for your comment. The participation rate of national health screening program was around 70%, thus despite of better socioeconomic status and health behaviors of screening examinees, the study population would cover most of the population. However, as your comment, socioeconomic inequalities in smoking prevalence in Korea has been reported [1] and smoking prevalence in this study population was lower than previous reports [1]. We reported this point as limitation of this study as follows;
“The current smoking rate in men and women in this study population was lower than that in the general Korean adult population [29,32]. However, considering the participation rate of the national health screening program (≥70%), the generalizability of their results could be applied to most of the Korean population.” (line 297-301)
“Finally, passive smoking and occupational and environmental exposures such as asbestos or radon exposures were not considered in the data set. However, to adjust for the effect of occupational and environmental exposures, we adjusted for the lung diseases as surrogate confounders that could be caused by occupational and environmental exposures.” (line 313-314, 318-320)
The authors note that they could not control for occupational and environmental exposures (particularly asbestos and radon) and that, instead, they adjusted for presence of lung diseases that could be caused by occupational and environmental exposures. I am not familiar with circumstances in Korea, but occupational and environmental exposures commonly vary by gender (the main focus of this study), SES (this sample is of higher SES than the general Korean population), and even smoking status. The authors do point out some potential for confounding but their concluding statement about this (lines 305-06) – that they adjusted for lung diseases associated with occupational and environmental exposures – does not feel to me a sufficient acknowledgement of the potential for confounding. Lung disease was reported in about 3% of men and 2% of women. This does not likely reflect adequate measurement of these other exposures of concern. How much of their findings do the authors think might be attributable to confounding?
Response: Thank you for your valuable comment. In Korea, the manufacture, import, supply and use of crocidolite, amosite and products containing these forms of asbestos were prohibited in 1997 and all kinds of asbestos were banned in 2009[2]. The estimated number of exposed population to asbestos were less than 200,000 annually between 1991 and 2003 among 50 million Korean population (<0.4%)[3]. The radon concentration in residential area was half the standard of Korean government or the United States Environmental Protection Agency[4]. The total number of asbestosis patients in Korean population for 16 years (between 1998 and 2013) were only 37[3]. Thus, we consider that occupational and environmental exposures would be very low and three lung diseases (chronic pulmonary obstructive disease, pneumoconiosis, and interstitial pulmonary disease) would be adequate as surrogate confounders to adjust for other occupational and environmental exposures. We pointed this as follows;
“The products containing asbestos have been prohibited partly since 1997 and completely since 2009 [34] and the proportion of workers exposed to asbestos was small (<1%) [35]. The radon concentration in residential area was half the standard of Korea and the United States [36]. Thus, we consider that occupational and environmental exposures would be very low in population. However, to adjust for the effect of occupational and environmental exposures, we adjusted for the lung diseases as surrogate confounders that could be caused by occupational and environmental exposures. In addition, other confounding variables such as cooking habits or diet could not be adjusted due to unavailable information.” (line 314-322)
Given they mention geographic variation in cancer rates by gender, their question “Does cigarette smoking influence the development of lung cancer and remain a key risk factor in women as in men?” has been raised on lines 53-54 seems like it should have a geographical component.
Response: Thank you for your valuable comment. We revised as followings
“Does cigarette smoking influence the development of lung cancer and remain a key risk factor in women as in men in East-Asian population?” (line 53-55)
One statement which appears twice (in the Simple Summary, lines 17-19, and Conclusions, lines 312-13) is confusing to this Reviewer: “However, higher lung cancer incidence in men than in women with the same level of smoking exposures was observed, suggesting a higher susceptibility for lung cancer in women.” Doesn’t this higher rate among men at the same level of exposure reflect greater susceptibility in men?
Response: We are sorry for confusion. As your comment, it means greater susceptibility in men. We revised the sentence as follows;
“~higher lung cancer incidence in men than in women with the same level of smoking exposures was observed, suggesting a higher susceptibility for lung cancer in men.” (line 17-19, 328-329)
There appears to be no mention of the figures until the figures actually appear at the end of the manuscript. The figures should either be presented explicitly in the body of the paper or be removed.
Response: Thank you for your comment. We presented the Figures in the manuscript as follows.
“Figure 1 and 2 presented cumulative lung cancer incidence in men and women according to smoking status and pack-years.” (line 154-155)
“Figure 3 presented cumulative lung cancer incidence according to years since cessation in former smokers.” (line 197-198)
The authors provide p-values for interaction terms between sex and smoking variables (for example on lines 160-61) but seem to do sex-stratified analyses throughout. How did the authors produce and evaluate interaction effects?
Response: We assessed interaction effect between sex and each smoking variable on lung cancer was analyzed using Cox regression models with interaction term. We added as follows in the Method
“The interaction effect between sex and each smoking variable on lung cancer risk was analyzed using Cox regression models with interaction term.” (line 122-123)
Another confusing statement is found on line 221-23: “A lower proportion of smoking among women with lung cancer (lower numerator) and a higher proportion of non-smokers among Asian women (higher denominator) possibly led to the lower incidence rate of lung cancer in female never smokers, especially in East Asia.” I think I understood their point given the sentence preceding this one, but it is difficult to follow. The authors talk about proportions in numerators and denominators instead of numbers which is confusing to people who expect numerators to be numbers and denominators to be person-time. I suggest rewording this sentence for clarity.
Response: Thank you for your comment. To clarify, we excluded the words lower numerator and higher denominator in the sentence and revised the sentence as follows;
“A lower proportion of smoking among women with lung cancer [6] and a higher proportion of non-smokers among Asian women [16] possibly led to the lower incidence rate of lung cancer in female never smokers, especially in East Asia.” (line 229-231)
Line 295 “showen” should be “shown.”
Response: We changed “showen” to “shown” (line 304)
Reference
- Chang, Y., et al., Long-term trends in smoking prevalence and its socioeconomic inequalities in Korea, 1992–2016. International Journal for Equity in Health, 2019. 18(1): p. 148.
- Kim, H.R., Overview of asbestos issues in Korea. Journal of Korean medical science, 2009. 24(3): p. 363-367.
- Kang, D.-M., et al., Occupational Burden of Asbestos-Related Diseases in Korea, 1998-2013: Asbestosis, Mesothelioma, Lung Cancer, Laryngeal Cancer, and Ovarian Cancer. Journal of Korean medical science, 2018. 33(35): p. e226-e226.
- Park, T.H., et al., Indoor radon concentration in Korea residential environments. Environmental Science and Pollution Research, 2018. 25(13): p. 12678-12685.
Reviewer 2 Report
This is a well-written manuscript presenting a carefully conducted analysis. Although the topic of smoking and lung cancer is (in my mind) over-studied, this remains one of the main public health problems.
The manuscript will benefit from the following addition:
- Individuals diagnosed with other than lung cancer at baseline should be included in the study population. The classical definition of the cohort study population is to include a population at risk of developing a specific disease. Thus, if an individual had colon cancer or melanoma, this individual is still at risk of lung cancer. In fact, this can be done as a sensitivity analysis.
- The quality of the figures should be better.
Author Response
Responses to the Reviewer 2’s Comments
Thank you for the valuable comments.
We have revised the paper in accordance with the comments, and the revisions are summarized below and highlighted in the manuscript.
This is a well-written manuscript presenting a carefully conducted analysis. Although the topic of smoking and lung cancer is (in my mind) over-studied, this remains one of the main public health problems.
The manuscript will benefit from the following addition:
- Individuals diagnosed with other than lung cancer at baseline should be included in the study population. The classical definition of the cohort study population is to include a population at risk of developing a specific disease. Thus, if an individual had colon cancer or melanoma, this individual is still at risk of lung cancer. In fact, this can be done as a sensitivity analysis.
Response: Thank you for your valuable comment. As your comment, we re-analyzed data including 45,952 people with other than lung cancer at baseline but the results were similar. We added this point in the Results section as follows;
“As a sensitivity analysis, we did same analysis including people having any other type of cancer except lung cancer before the date of examination.” (line 129-130)
“Sensitivity analysis including people with other than lung cancer before the date of examination showed comparable results.” (line 209-210)
- The quality of the figures should be better.
Response: We are sorry for low resolution of figures. To improve the quality of figures, we re-uploaded image files instead of ppt file.
Reviewer 3 Report
This is a large epidemiological study of factors associated with lung cancer in a korean population. It is well written overall The main conclusion of the study is that women had a lower risk of developping lung cancer than men with comparable smoking status. It seems that as the authors suggest in their introduction, epidemiological studies are not unanimous as to whether women are less or more at risk of developping lung cancer at an equal smoking status as men. It seems clear that depending on the studied population, the results may be different.
The strength of this study is that it concerns a very large number of subjects. The external validity of its conclusion are probably high when applied to the Korean population. The major weakpoint is, obviously, the lack of explanation of this observed difference. Especially that the authors are unable with the available date to adjust their analysis to known risk factors especially in the asian population such as diet, cooking habits, occupational and environmental exposures...
This sentence (ln 17-18) needs to be revised : " However, higher lung cancer incidence in men than in women with the same level of smoking exposures was observed, suggesting a higher susceptibility for lung cancer in women."
Author Response
Responses to the Reviewer 3’s Comments
Thank you for the valuable comments.
We have revised the paper in accordance with the comments, and the revisions are summarized below and highlighted in the manuscript.
This is a large epidemiological study of factors associated with lung cancer in a korean population. It is well written overall The main conclusion of the study is that women had a lower risk of developping lung cancer than men with comparable smoking status. It seems that as the authors suggest in their introduction, epidemiological studies are not unanimous as to whether women are less or more at risk of developping lung cancer at an equal smoking status as men. It seems clear that depending on the studied population, the results may be different.
The strength of this study is that it concerns a very large number of subjects. The external validity of its conclusion are probably high when applied to the Korean population. The major weakpoint is, obviously, the lack of explanation of this observed difference. Especially that the authors are unable with the available date to adjust their analysis to known risk factors especially in the asian population such as diet, cooking habits, occupational and environmental exposures...
Response: Thank you for your comment. As your comment, we only adjusted for age, alcohol consumption, exercise, body mass index, family history of cancer and three lung diseases due to limited available variables. Instead, we considered lung diseases confounders that could be caused by occupational and environmental exposures as surrogate confounders. However, we could not adjust for other variables such as diet or cooking habits. We included it as one of our limitations as follows;
“Finally, passive smoking and occupational and environmental exposures such as asbestos or radon exposures were not considered in the data set. The products containing asbestos have been prohibited partly since 1997 and completely since 2009 [34] and the proportion of workers exposed to asbestos was small (<1%) [35]. The radon concentration in residential area was half the standard of Korea and the United States [36]. Thus, we consider that occupational and environmental exposures would be very low in population. However, to adjust for the effect of occupational and environmental exposures, we adjusted for the lung diseases as surrogate confounders that could be caused by occupational and environmental exposures. In addition, other confounding variables such as cooking habits or diet could not be adjusted due to unavailable information.” (line 313-322)
This sentence (ln 17-18) needs to be revised : " However, higher lung cancer incidence in men than in women with the same level of smoking exposures was observed, suggesting a higher susceptibility for lung cancer in women."
Response: We are sorry for confusion. As your comment, it means greater susceptibility in men. We revised the sentence as follows;
“~higher lung cancer incidence in men than in women with the same level of smoking exposures was observed, suggesting a higher susceptibility for lung cancer in men.” (line 17-19, 328-329)